# Boar Semen Contamination: Identification of Gram-Negative Bacteria and Antimicrobial Resistance Profile

**DOI:** 10.3390/ani12010043

**Published:** 2021-12-27

**Authors:** Luminita Costinar, Viorel Herman, Elena Pitoiu, Ionica Iancu, Janos Degi, Anca Hulea, Corina Pascu

**Affiliations:** 1Infectious Diseases Department, Faculty of Veterinary Medicine, Banat’s University of Agricultural Sciences and Veterinary Medicine “King Michael I of Romania” from Timisoara, 119 Aradului Street Timisoara, Timis County, 300645 Timisoara, Romania; luminita.costinar@usab-tm.ro (L.C.); viorel.herman@fmvt.ro (V.H.); ionica.iancu@usab-tm.ro (I.I.); janosdegi@usab-tm.ro (J.D.); anca.hulea@usab-tm.ro (A.H.); 2Synevovet Laboratory, Industriilor Street, No. 25, Chiajna, Ilfov County, 077040 Chiajna, Romania; elena.pitoiu@synevo.com

**Keywords:** boar, semen microbiota, antimicrobial resistance profile

## Abstract

**Simple Summary:**

Boar semen can contain many bacterial species, some of which can have a negative impact upon the quality of the semen, as well as on the sows’ reproductive capacity. Semen contamination may occur at time of collection or during semen processing. The aim of this study was to identify gram-negative bacteria that appear in boar semen and to establish models of antimicrobial resistance of isolated gram-negative bacteria. Semen doses examined contained bacterial species with a known negative effect on sows’ reproductive tracts (*Pseudomonas, Enterobacter, Klebsiella, E. coli*), and more than half of these isolates were resistance to gentamycin (56.52%) and penicillin (58.69%) antimicrobials commonly used in boar semen extenders. This work proved the presence of pathogenic multiple resistant bacteria in semen, and therefore, we highly recommend periodic microbiological screening of bacteriospermia in boars to avoid the use of low-quality semen in the pig industry.

**Abstract:**

Bacterial contamination of boar semen occurs with some frequency in artificial insemination centers and may have a negative effect on the quality of the semen as well as on the sows’ reproductive capacity. Normally, the source of bacterial contamination in pig seminal doses is the own boar. However, distilled water or laboratory equipment used to elaborate the seminal doses can be an important source of bacterial contamination. This study focused on the identification of gram-negative bacteria in boar semen, and impact on the quality of ejaculates obtained from boar, as well as on the establishment of antimicrobial resistance patterns of isolated gram-negative bacteria. Semen samples were collected from 96 boars, ranging in age from 12–36 month, from three artificial insemination centers from the North-West of Romania. Bacterial species were identified by two methods: matrix-assisted laser desorption/ionization time-of-flight (MALDI-TOF) mass spectrometry and API 20 E (BioMerieux, France). The main bacteria isolated from the doses diluted semen were gram-negative bacteria (47.91%), with a majority of the contaminant bacteria belonging to the family *Enterobacteriaceae*: *Seratia marcescens* 19.56%, *Proteus mirabilis* 15.21% and *Escherichia coli* 10.86% and to the family *Pseudomonaceae*: *Ralstonia picketii* 17.39%, *Burkholderia cepacia* 10.86%, *Pseudomonas aeruginosa* 8.69%, and *Pseudomonas fluorescens* 4.34%, respectively. More than half of these isolates (56.52%) were resistant to gentamycin and 58.69% were resistant to penicillin. These antibiotics are very frequently added in sperm diluent in the centers for the processing of sperm from boars in Romania. Regular monitoring for bacterial contamination is an important aspect of a control program.

## 1. Introduction

In the pig breeding industry, diluting and preserving the longevity of semen greatly improves and streamlines artificial insemination techniques. Through the techniques of dilution and preservation of sperm life, other microorganisms such as bacteria, fungi can adversely affect the quality of diluted semen. Bacterial contamination of semen or bacteriospermia is a fairly widespread problem in semen collection and processing centers [1,2]. For this reason, antimicrobial substances are introduced in the diluted doses of semen, substances that prevent the development and multiplication of these bacteria. These antimicrobial substances are chosen on the basis of efficacy on the main bacteria isolated and identified in the doses of diluted semen. These bacteria are usually gram-negative bacteria.

The main sources of bacterial contamination that influence the initial number of bacteria in the semen are boar feces, prepuce, preputial fluids, prespermatic fraction, hair, boar skin, boar’s respiratory secretions, people (laboratory workers, visitors), drinking water, distilled water used in the preparation of diluted doses of semen, feed, shelter, air, ventilation system, testicular, urethral, bladder infections, laboratory materials (peristaltic pump, boxes, BPS station, etc.) [3,4,5]. Overgrowth by contaminant bacteria of certain genera has a deleterious effect on semen quality and longevity [5]. Agglutination of sperm occurs, decreases sperm motility and viability and as a result sows will present regularly reproductive disorders translated by estrus returns, postinsemination vulvar discharges, abortions, mummifications, and the low reproductive performance of the herds were reported [4,5].

Strict and rigorous attention to hygiene during semen collection and processing may reduce bacterial contamination [5]. The normal flora of the skin, hair and respiratory tract of boar cannot be reduced. However, personnel can minimize the bacterial load by monitoring as strictly as possible the rules of hygiene and sterility of the collection and processing equipment of the semen. It is particularly important to know the bacterial microbiota in boar semen and the profile of their antimicrobial resistance [6,7,8].

The objectives of present study focused on identification and microbiological examination of the boar sperm doses and analysis of the quality of these samples, using a rapid and precise working technique, the matrix-assisted laser desorption/ionization time-of-flight (MALDI-TOF) mass spectrometry and API 20 E (BioMerieux, Marcy l᾽Etoile, France) for strains identification and further the establishment of the main antimicrobial resistance pathotypes of isolates gram-negative bacteria.

## 2. Materials and Methods

A 1-year (April 2020–April 2021) retrospective study was performed by sending 96 extended swine semen for routine quality control bacteriological screening at the Banat University of Agricultural Sciences and Veterinary Medicine “King Michael I of Romania”, Timisoara, Romania, Department of Infectious Diseases.

### 2.1. Origin of Samples

Semen samples were collected from 96 boars, ranging in age from 12 to 36 months, from 3 artificial insemination centers located in northwestern Romania. The boars included in this study belonged to different breeds (Large White, Landrace, Duroc, Pietrain, and PIC).

Semen doses from these boar centers were distributed in 46 pig farms.

Eighty boars from the first center belonged to Large White, Landrace breed, and the hybrid PIC. Form this center there were collected 44 raw semen samples. Sixty-four boars from the second center belonged to Large White, Duroc, and Pietrain breed, and there were collected 35 raw semen samples. Thirty boars from the third center belonged to Duroc and Large White breed, and there were collected 17 raw semen samples. The number of samples was chosen so as to be representative for each boar center. All centers used the same extender (gentamicin sulfate, 12.5 g) which contain gentamicin as antibiotic.

All the boars were routinely used for artificial insemination (AI), received a commercial feed pellet and were housed in individual boxes equipped with nipple drinkers, according to the European Commission Directive for Pigs Welfare.

### 2.2. Semen Evaluation

All boars were clinically healthy, but on farms there were registered reproductive failures manifested by genital and urinary tract infections, abortions, etc. Collection of ejaculate was performed by manual pressure, after the sanitization of the prepuce zona. The prespermatic fraction of the raw ejaculations was discarded to maintain only the sperm-rich fractions.

After collecting the semen samples, the volume, concentration, color, and motility were evaluated at their own board stud where the samples came from. Color was assessed visually by observing the semen in transparent microtubes, examining the degree of turbidity, presence of the blood or an unusual color.

The sperm volume was assessed immediately after the ejaculate sampling, by direct observation in the graduate sampling container.

The sperm concentration was also determined immediately after the ejaculate was collected, using the Sperm Sue spectrophotometer.

The semen analysis (concentration and mobility) was performed using the CASA IVOS version 12 system produced by Hamilton–Thorne Bioscience, using Animal Motility Software, Viadent option. For the sample analysis, Leja blades of 30 µL, with 4 chambers (Cryo BioSystem, France) were used, in which 10 µL semen was placed, with the help of an automatic pipette and subsequently the samples were analyzed. The system scanned automatically 10 different microscopic fields through the chamber.

After raw semen samples examination, all 96 samples were diluted 1:9 with a standard commercially extender which contains gentamycin sulfate and glucose. An aliquot of 200 µL of each diluted semen sample was added to sterile microtubes containing Stuart transport medium and transported at Faculty of Veterinary Medicine for bacteriological exams. The raw semen and the extender were bacteriologically tested in the boar centers laboratories.

### 2.3. Bacterial Isolation and Identification

In this study, culture was performed by inoculating aliquots of diluted semen (DS) on the surface of Columbia blood agar (Oxoid, Hampsire, UK) with a sterile glass L shape hockey stick spreader. The Petri dishes were incubated under aerobic conditions at 37 °C and after 24–48 h each morphologically different colony was used for subcultivation on Columbia blood agar, Mac Conkey agar (Oxoid, Hampsire, UK) and EMBL (Eosin Methylene Blue) agar (Oxoid, Hampsire, UK).

After cultivation on specific media the cultures were examined every 6 h. Isolated bacteria were identified using standard microbiological procedures: growth and colonial characteristics, gram staining, cellular morphology, catalase, and oxidase reaction, hemolysin production and coagulase test.

The final pure culture represented a basic material for bacterial identification by API 20 E system (bioMerieux, France) and MALDI-TOF, with MALDI Biotyper (Bruker Daltonic, Karlsruhe, Germany).

### 2.4. Antimicrobial Susceptibility Test

Susceptibility tests were performed using disc diffusion method (Kirby–Bauer), according to the Clinical and Laboratory Standard Institute (CLSI) protocol [9,10]. As there are no CLSI susceptibility breakpoints available for *Ralstonia pickettii* or *Burkholderia cepacia*, the antibiotic susceptibility results were interpreted using the CLSI criteria for *Pseudomonas* spp. For this purpose, the antibiotics most often used in reproductive diseases in sows were used. The antimicrobial agents tested included: ceftiofur (30 µg), lincomycin (2 µg), enrofloxacin (5 µg), gentamycin (10 µg), neomycin (30 µg), flumequine (30 µg), apramycin (30 µg), penicillin (10 IU), and ampicillin (10 µg). As the control strain was used *Escherichia coli* ATCC 25922.

### 2.5. Statistical Analyses

The data were processed using the nonparametric test Mann–Whitney U, to assess the difference in semen concentration and motility in regard to bacterial isolation.

A coefficient of variation (CV%) was calculated to analyze the variability between the boar centers.

The cluster examination for antimicrobial resistance was performed with BIONUMERICS 8 (Applied Maths, a bioMerieux company). The results were obtained by a temporary evaluation license, and we have received the permission to publish these results.

## 3. Results

No alterations regarding sperm color and concentration were observed, but the presence in semen samples of *E. coli, Burkholderia cepacia, Serratia marcescens,* and *Proteus mirabilis* was negatively associated with sperm motility (*p* < 0.05).

The boar ejaculate is milky white in all raw semen samples with no other shade or color, and the presence of blood was not noticed. Normal color of boar ejaculate is white, with bluish shadows [1,2].

Regarding the concentrations of the harvested semen, following the application of the nonparametric Mann–Whitney U test, there were no distinctly significant differences (*p*
˃ 0.05) between boar breeds and boar centers.

The mean concentration and motility of sperm cells were 389 ± 127.5 (×10^6^ mL^−1^) and 91.9 ± 7.3% respectively.

There were 21 positive samples (47.72%) for bacterial contamination from the first center, 16 positive samples (45.71%) from the second center, and 9 positive samples from the third center (52.94%).

Statistical analysis of variability between the artificial insemination centers was 39.31%, meaning a low degree of dispersion of values and no difference between boar centers.

Out of the 96 tested doses of diluted semen (DS), only 46 (47.91%) were positive at bacteriological exams, the other 50 samples were microbiologically negative. From the bacteriologically positive semen samples, only 6 samples (13.04%) presented mixed contamination, with more than one bacterial species being isolated.

Aerobic cultivation performed on 96 doses of diluted semen (DS) led to the isolation of 9 different species of bacteria identified through MALDI-TOF and API 20 E (Bio Merieux, Marcy l᾽Etoile, France) as pathogen bacteria, as well as skin and mucosal commensals and environmental bacteria. The specific species were *Serratia marcescens, Ralstonia pickettii, Proteus vulgaris, Pseudomonas fluorescens, Burkholderia cepacia, Klebsiella oxytoca, Pseudomonas aeruginosa, Enterobacter* spp. and *Escherichia coli* (Table 1, Figure 1). The most frequently occurring microorganisms were represented by *Serratia*
*marcescens*, Ralstonia* pickettii *and Proteus* mirabilis.*

In addition to these gram-negative bacteria with known pathogenic action, gram-positive bacteria were also identified as *Streptococcus porcinus, Staphylococcus equorum, Staphylococcus succinus,* and *Aerococcus viridans*. These isolates were not further tested because gram-negative bacteria have the most harmful effect on the semen [3,4].

The antimicrobial susceptibility characterization was performed in all 46 isolates obtained (Table 2).

The resistance profiles cluster analysis resulted in three groups. The first group included 13 isolates (from a total of 21 isolates) which belong to the *Pseudomonaceae* family. The second group included 23 isolates with heterogeneous resistance profile, including 18 isolates from the *Enterobacteriaceae* family. The third group included two isolates of *Proteus mirabilis* and *E. coli* with different behaviors to antimicrobials (Figure 2).

## 4. Discussion

The sources of bacterial contamination of semen doses are many and very diverse. Althouse and Lu (2005), in the studies performed, described the bacterial strains belonging to 25 different genera that have already been detected as semen contaminants. The presence of bacterial contamination in pig semen doses may have a negative effect on their quality and durability [3].

Of the 96 doses of diluted semen (DS) studied, only in 46 doses (47.91%), the presence of bacteria was demonstrated. This relatively high percentage of positivity demonstrates that bacteriospermia or bacterial contamination is frequent in laboratories and semen processing centers. Beneman et al. (2018), and other researchers reported even higher bacterial contamination percentage (86%) [7,11]. Of the total samples tested (96 samples), 50 samples were microbiologically negative.

According to the data presented in Table 1 and Figure 1, several genera and gram-negative bacterial species were isolated from the diluted seminal material. Bacterial species were present in 46 samples of the 96 doses of doses of diluted semen studied (47.91%). From 31 samples (67.39%), a single gram-negative bacterial genus was isolated. In the other 15 samples (32.60%), two gram-negative bacterial genera were identified.

Serratia marcescens was the most frequent isolated (19.56%), followed by *Ralstonia pickettii* (17.39%), *Proteus vulgaris* (15.21%), *E. coli* (10.86%), *Burkholderia cepacia* (10.86%), *Klebsiella oxytoca* (8.69%), *Pseudomonas aeruginosa* (8.69%), *Enterobacter* spp. (4.34%), and *Pseudomonas fluorescens* (4.34%). Martin et al. (2010) identified gram-negative bacteria, the most common isolated microorganism being *E. coli* (79%), followed by *Proteus* spp. (36%), and *Pseudomonas* spp. (8%), and but also gram-positive bacteria—*Staphylococcus* spp. (12%), and *Streptococcus* spp. (9%) [12].

In a study conducted by Tvrda et col. [13] in Slovakia, 12 bacterial genera and 16 bacterial species were isolated and identified in boar ejaculates immediately following semen dilution, using MALDI-TOF mass spectrometry, as follows: *Proteus vulgaris, E. coli, Pseudomonas aeruginosa, Pseudomonas putida, Klebsiella pneumoniae, Aerococcus viridans, Staphylococcus aureus, Staphylococcus chromogenes, Staphylococcus simulans, Clostridium difficile, Enterococcus hirae, Bacillus cereus, Bacillus licheniformis, Bacillus subtilis, Acinetobacter iwoffii, Rothia nasimurium,* and *Corynebacterium* spp.

In our research, *Serratia marcescens* was isolated in a proportion of 19.56%. In other studies, the proportion in which *S. marcescens* was isolated is variable. Althouse et al. [2] isolated *S. marcescens* in proportion of 10.3%, Ubeda et al. [1] in proportion of 12.5%. Schultze et al. [14] isolated *S. marcescens* in proportion of 2.3% and *Ralstonia pickettii* in proportion of 11.4%.

Most of the bacteria isolated and identified in this study are opportunistic bacteria, but which can form, alone or in combination with other bacteria, biofilm on the surfaces in the semen processing laboratory [15,16]. For this reason, we consider that it is particularly important to have a high degree of hygiene of the staff, equipment, and laboratory where the semen is taken and processed.

Tvrda et al. [13] reported that 76% of semen samples had been contaminated with relatively high variety of bacterial genera, predominantly well-known uropathogens.

*Serratia marcescens* is a bacterial contaminant that can be spermicidal when is present in extended boar semen [15,16,17]. This particular contaminant appears to originate from carrier boars, where it resides in the preputial cavity, but has also been shown to easily contaminate the semen-processing laboratory. Regarding the *Serratia marcescens* it was observed its high capacity to create biofilm on wet surfaces and deteriorate the sperm cell to the point of causing sperm death quickly.

*Ralstonia pickettii*, another gram-negative, nonfermentative bacteria, was isolated in this study in a proportion of 17.39%. These bacteria can be isolated from the system of producing distilled water; that is, water used for diluting semen [16]. From a medical point of view, these strains are of particular importance, because they are responsible for the appearance of the pyometra in sows, after insemination. Similar results were obtained by other researchers [4,7,18,19,20,21,22]. *Ralstonia picketii* and *Achromobacter xylosoxidans* can be found in the water distillation system of a boar stud facility that uses this water to expand raw semen. Clark et al. [16] and other research showed that the presence of *A. xylosoxidans* and *R. pickettii* in water for semen extension of porcine semen does not detrimentally affect sperm motility or pH of the final solution regardless the choice of semen diluent [12,16,18,23,24].

*Escherichia coli* was isolated in proportion of 10.86%. Our results differ from the findings of Maroto Martín et al. [12]. They reported the presence of *E. coli* in a proportion of 79%, followed by *Proteus spp*. 36% and *Pseudomonas spp*. 8%. In Italy, other researchers isolated *E. coli* 53% [23]. The results obtained in this study are similar to the results obtained by Althouse et al. [3], research where *E. coli* was isolated in proportion of 6.4%.

Enteric bacteria, especially *E. coli,* negatively affect fertility by decreasing sperm motility, affecting the acrosome, and causing sperm agglutination [12,14,25,26,27,28,29]. In research conducted by Ubeda et al. (2013), *Klebsiella oxytoca* was isolated in proportion of 11.79%, *Serratia marcescens* 12.55%, and *Escherichia coli* 1.52% [1].

Maroto Martín [12] and Martins [27] confirm the general opinion that boar ejaculates are more predisposed to gram-negative bacteria contamination.

Semen contamination becomes relevant when it is associated with reduction of male fertility or with decreased semen boar quality. Bacterial contamination of the sow᾽s reproductive tract by artificial insemination can cause metritis, endometritis, vulvar discarches, return to the estrus, reduction of litter size, and an increased number of stillbirths and mummies [14,17,18,21,30,31,32,33]. Although the boars did not have clinical symptoms, we cannot say that they did not have bacteriospermia, since many infections of the male reproductive tract do not imply clinical diseases and are only visible when a poor seminal quality is perceived. As such, reproductive indices decrease or females increasingly present metritis, uterine infection, etc. For this reason, we consider that it is particularly important to have a high degree of hygiene among the staff, equipment, and laboratory where the semen is processed. The production of semen doses with low bacterial contamination and high viability of sperm will be possible only with a strict hygienic control in the processing of semen and having the antibiotic resistance profile of bacteria that can contaminate semen [12,34,35,36].

The antimicrobial usage in semen extenders aims to reduce the bacterial contamination. However, it is known that 80–90% of the bacteria isolated from the doses of semen present various levels of antibiotic resistance [12,21,37,38]. In this study, all gram-negative bacterial strains presented resistance phenotype against at least one of the tested antimicrobial groups.

The extender used in the artificial insemination center contains gentamycin. In our study 26/46 isolates (56.52%) showed resistance to gentamycin, 11 were intermediate and 10 isolated were sensitive to this antibiotic. This shows that the effectiveness of the antibiotic in doses of diluted semen is quite low. The behavior of bacterial strains for gentamycin can be different; other researchers [7,28] obtained 80% of isolates sensitive to gentamycin [7].

In their study, Tvrda et al. [13] reported that gentamycin was effective enough to eradicate gram-negative bacteria*, E. coli, Klebsiella pneumoniae, Proteus vulgaris* strains showed 100% sensitivity to ampicillin. We can according to Maroto Martín et al. [12] Bresciani et al. [25] and Gączarzewicz et al. [28], who pointed out that a significant proportion of the bacteria commonly found in boars ejaculate in Europe may be resistant to gentamicin.

Most of the gram-negative bacteria isolated in our study had a high-level resistance rate of antimicrobials when tested against neomycin, penicillin, lincomycin, and ceftiofur (Table 1). These results are correlated with previous studies [13,31,34] reporting that several bacterial genera and species exhibit a certain degree of resistance to gentamycin and aminoglycosides (the most common antibiotics used in semen extenders) leading to a concerning assumption that none of these antibiotics was able to effectively eradicate Gram negative bacteria present in diluted semen samples.

## 5. Conclusions

In this study, following the bacteriological examinations of the doses of diluted semen, 47.91% of them were positive, demonstrating high bacterial contamination. A predominance of gram-negative opportunistic bacteria were observed in the contaminated samples, which may be involved in uterine infections in sows or in reducing the number of fetuses.

We consider it essential to make an accurate bacterial-type diagnostic and quantification method, as well as a proper antibiotic selection to use for the dose’s diluent.

This work proved the presence of pathogenic gram-negative bacteria with multiple resistance to antibiotics in semen, and therefore, we highly recommend periodic microbiological screening of bacterial contamination in boars to avoid the use of low-quality semen in the pig industry.

Hygienic semen collection and processing techniques and stringent laboratory procedures must be the first and primary lines of defense in successfully managing contamination. Controlling bacterial growth in extended semen with antibiotics must be a secondary method of bacterial contamination of the doses of diluted semen.

Semen processing is not yet standardized among artificial insemination centers from Romania, and the critical points during production need to be identification.

## Figures and Tables

**Figure 1 animals-12-00043-f001:**
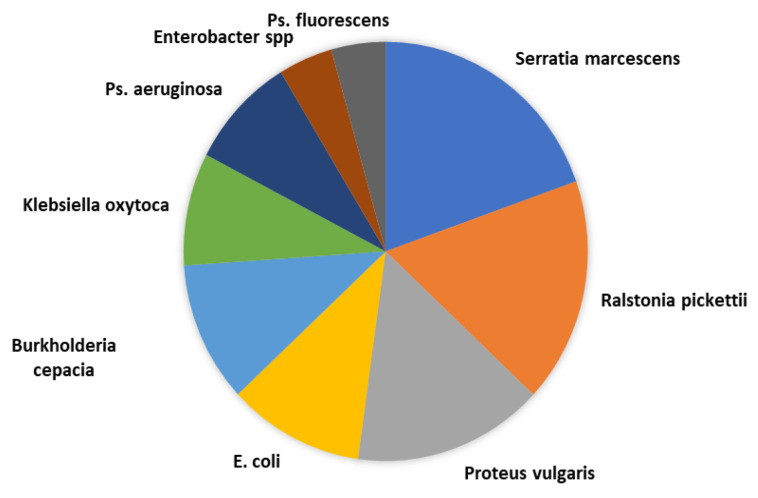
Proportion of gram-negative bacteria isolated and identified in doses of diluted semen (DS).

**Figure 2 animals-12-00043-f002:**
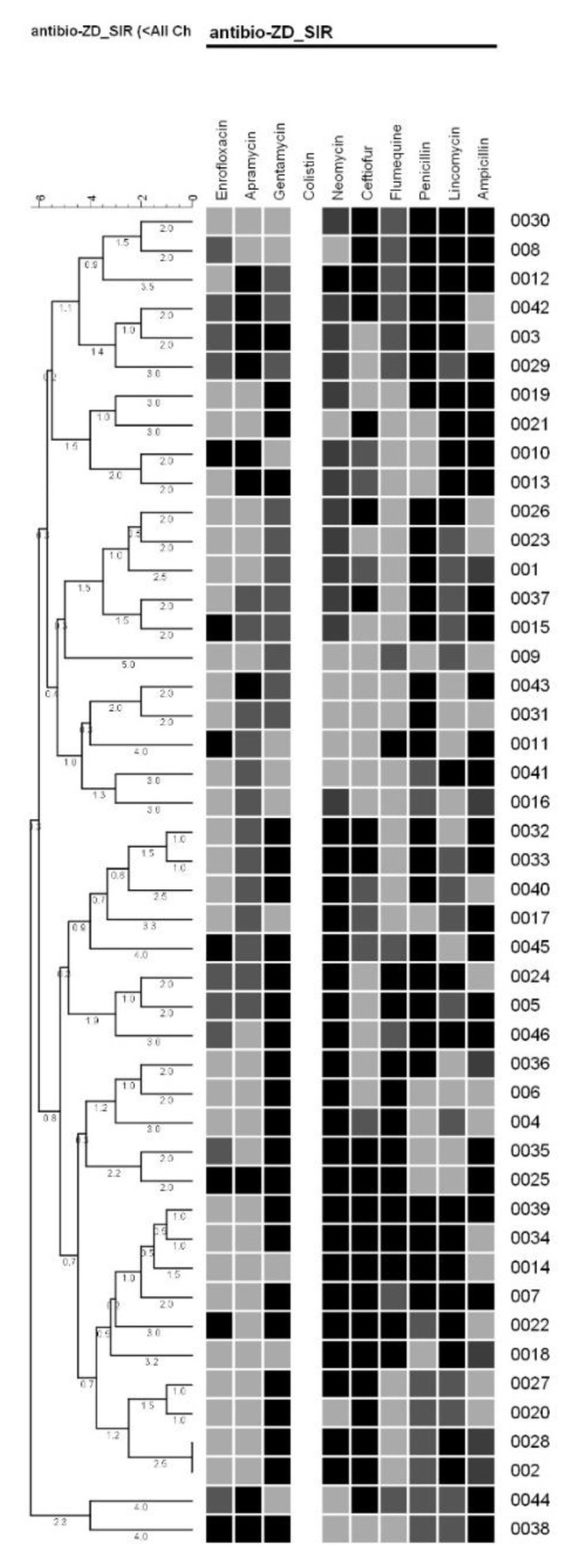
Resistance profile cluster analysis of gram-negative isolates. Black, gray, and white colors correspond to resistance, intermediate, and sensitivity to antimicrobials, respectively.

**Table 1 animals-12-00043-t001:** Synopsis of gram-negative bacterial genera and species isolated from doses of diluted boar semen (DS) isolated in pure culture or in mixed culture.

	Positive Semen Samples	
Bacterial Genera and Species	No	%
*Serratia marcescens*	9	19.56
*Ralstonia pickettii*	8	17.39
*Proteus mirabilis*	7	15.21
*Escherichia coli*	5	10.86
*Burkholderia cepacian*	5	10.86
*Klebsiella oxytoca*	4	8.69
*Pseudomonas aeruginosa*	4	8.69
*Enterobacter* spp.	2	4.34
*Pseudomonas fluorescens*	2	4.34
Total	46	100

**Table 2 animals-12-00043-t002:** Behavior of isolated gram-negative bacterial strains in antimicrobial substances.

Antimicrobial Substance	Number of Strains		
	Sensitive	Intermediate	Resistance
Enrofloxacin 5 µg	29	10	7
Apramycin 30 µg	21	14	10
Gentamycin 10 µg	10	10	26
Neomycin 30 µg	8	11	27
Ceftiofur 30 µg	17	7	22
Flumequine 30 µg	22	11	13
Penicillin 10 IU	11	9	26
Lincomycin 2 µg	10	14	22
Ampicillin 10 µg	15	8	23

IU: International Units.

## Data Availability

The data presented in this study are available on request from the corresponding author.

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
