# Peer review of "Boar Semen Contamination: Identification of Gram-Negative Bacteria and Antimicrobial Resistance Profile"

_animals, 2021, doi:10.3390/ani12010043_

Round 1
Reviewer 1 Report
Dear Authors,
Manuscript can be published in Animals.
Author Response
Dear Editor,
We would like to thank the editor and each reviewer for their interest, positive comments, and thoughtful replies to our manuscript. We are delighted to be considered for publication in Animals, and to receive quality peer reviews that have greatly improved our manuscript. We have thoroughly considered each Reviewer's comments and have revised the text to reflect this. We have addressed each Reviewer’s comments below, in a point by point fashion, as requested. The new version of the manuscript contains the requested changes and additional information (as reviewers required). Below, you can find the answers to the all comments and suggestions indicated with red font.
Thank you again for your time and consideration.
Sincerely yours,
Dr. Luminita Costinar
On behalf of all the authors
Reviewer Comments:
Reviewer #2:
Review Manuscript ID: animals-1425121, entitled ‘Boar semen microbiota: identification of bacterial strains and antimicrobial resistance profile’
COMMENT 18: In my opinion, this work cannot be presented as a study of the Boar semen microbiota, since the semen was cultured diluted in extenders with antibiotics.
Boar semen contamination: identification of bacterial strains and antimicrobial resistance profile
The authors have modified the Title, I agree
COMMENT 19: In Material and Methods affirm “a 200 μl aliquot of each semen sample was added to sterile micro-88 tubes containing Stuart transport medium” (lines 88-89). What implies that the pure semen is the one that is cultured, however, in the discussion and in the conclusion it is described that the semen has been cultured diluted in extender with antibiotics.
The authors have clarified this issue
COMMENT 20: (Lines 234-242) “If it is taken into account that these strains were isolated from the diluted semen, given that the diluent contained two antibiotics and the boars were clinically healthy, it can be hypothesized that the bacteriospermia found is the consequence of semen contamination during harvesting or processing semen processing laboratory. For this reason, we consider that it is particularly important to have a high degree of hygiene of the staff, equipment and laboratory where the semen is taken and processed. The production of doses of semen with low bacterial contamination and high viability of sperm will be possible only with a strict hygienic control in the processing of semen, and with the knowledge of the antibiotic resistance profile of bacteria that can contaminate semen.”
The authors have partially clarified this issue
New comments:
1.- The authors comment that “the study is carried out with semen samples were collected from boars clinically healthy, but reproductive failure manifested by genital and urinary tract infections, abortions, etc. were recorded in the farm”. However, in the study they decided to analyse only gram - bacteria, because gram – bacteria have the most harmful effect on the semen.
In my opinion, they should have analysed all microorganisms, as in most of the bibliography referred.
In addition to these gram-negative bacteria with known pathogenic action, non- pathogenic gram-positive bacteria were also identified - Streptococcus porcinus, Staphylococcus equorum, Staphylococcus succinus and Aerococcus viridans. These isolates were not further tested because it is known from the literature that gram-negative bacteria have the most harmful effect on the semen (lines 222-226)
Especially when the authors comment on the importance of semen contamination not only in the quality of the semen but also its possible pathogenicity in the reproductive tract of the sow.
The bacterial contamination of the sow reproductive tract by insemination artificial can cause metritis, endometritis, vulvar discarches, return to the oestrus, reduction of litter size, increased number of stillbirths and mummies. (lines 316-319)
In fact, the pathogenicity of some gram + bacteria in different pathological processes in the sow has been described. For example, some gram+ bacterias isolated in the study have been described in the following publications:
- Wang, Y., Guo, H., Bai, Y. et al. Isolation and characteristics of multi-drug resistant Streptococcus porcinus from the vaginal secretions of sow with endometritis. BMC Vet Res 16, 146 (2020). https://doi.org/10.1186/s12917-020-02365-9
A S. porcinus isolate with multi-drug resistance was identified from vaginal secretions of sows with endometritis in one pig breeding farm, which suggests that the sow endometritis was caused by S. porcinus infection during artificial insemination. This study indicates that sensitive antibiotics such as penicillin G or some cephalosporins could be used for treatment of the diseases. In addition, the study hints that bacterial multi-drug resistance is a tough problem for disease treatment in pig farms.
- Nguyen VG, Kim CU, Do HQ, Shin S, Jang KC, Park YH, Park BK, Chung HC. Characteristics of Aerococcus viridans isolated from porcine fetuses in Korean farms. Vet Med Sci. 2021 Jul;7(4):1325-1331. doi: 10.1002/vms3.456. Epub 2021 Feb 24. PMID: 33624943; PMCID: PMC8294361.
Swine abortion caused by viruses as well as bacteria has caused many economic losses in domestic farms over the years; however, bacterial abortion has not yet been studied in Korea. Several bacterial species were isolated from aborted fetuses (n = 103) for which the cause of death was not viral abortion. Among them, we focused on Aerococcus viridans, which had the highest positive rate within three provinces (Gangwon, Jeonnam and Gyeongnam). A total of 16 isolates were identified as A. viridans by matrix-assisted laser desorption ionization-time of flight mass spectrometry (MALDI-TOF MS), and 13 were characterized by both antibiotic resistance and 16S rRNA gene analysis. Based on antibiotic susceptibility testing result, eight antimicrobials could not effectively eliminate the present isolation (more than 40% of isolates can resist these antibiotics), while all except two strains were susceptible to trimethoprim/sulfamethoxazole. Molecular analysis indicated genetic variation among these strains. This study is the first report detecting A. viridans from aborted fetuses in Korean domestic farms.
Response new comments 1: Thank you for taking the time to review our manuscript and for valuable comments and suggestions, which have helped us to significantly improve its quality. We fully agree with your comment and thank you in particular for this suggestion about including in this study only gram – bacteria.
To correct this, we will modify the article`s title as: “Boar semen contamination: identification of gram-negative bacteria and antimicrobial resistance profile” (lines3-4). In future studies we will also include gram-positive bacteria (Streptococcus spp., Aerococcus, viridans, Staphylococcus spp. etc.).
Special thanks for this suggestion. We completely agree your observation.
In future studies we will also include gram-positive bacteria (Streptococcus spp., Aerococcus, viridans, Staphylococcus spp. etc.).
Thank you again!
2.- On several lines it is commented that gram- bacteria have only been found when the lines 222-226 specify the isolation of several gram + bacteria.
All isolates are gram-negative bacteria, and no any gram-positive bacteria were identified (line 219)
According to the data presented in Table 1 and Figure 1 several genera and bacterial species were isolated from the diluted seminal material, all of them being Gram negative bacteria (lines 267-268)
Response new comments 2: Thank you for this correction. In lines 222-226, we wanted to clarify the fact that, in addition to gram-negative bacteria, gram-positive bacteria were also isolated, but these were not taken into account in the present study.
3.-The contamination of swine semen is a general concern that generates frequent publications, so it is important to compare the most recent publications in the discussion of the articles, a more recent and more in-depth bibliographic review on the subject has been missed.
For example:
Andressa C. Dalmutta,b, Luisa Z. Moreno b , Vasco T. M. Gomesb , Marcos P. V. Cunha c , Mikaela R. F. Barbosad , Maria Inês Z. Satod , Terezinha Knöblc , Antonio Carlos Pedrosoa and Andrea M. Moreno b Characterization of bacterial contaminants of boar semen: identification by MALDITOF mass spectrometry and antimicrobial susceptibility profiling JOURNAL OF APPLIED ANIMAL RESEARCH 2020, VOL. 48, NO. 1, 559–565 https://doi.org/10.1080/09712119.2020.1848845
Response new comments 3: We fully agree with the suggested thus new titles were introduced in the bibliography and the text was improved in the discussions.
We agree the addition of this representative reference. Thus, the reference was added (see the reference no. 13, 21, 28, 30, 35 from the reference list of the revised version)
4.- The discussion not contributes new information to the topic.
Response new comments 4: Thank you very much for your valuable advices for improvement of the manuscript. We consider that for our country these data obtained in this study bring new information, taking into account that there are not enough studies carried out in Romania on this subject. Ciornei et al. have a study of sperm bacteria in pigs (27).
5.- I am not an expert in any English language, but basic inaccuracies of the language are appreciated
Response new comments 5: We fully agree with this requirement and have tried to comply with and improve it.
In conclusion, although the authors have managed to give a new focus to the article, the selection of the bacteria under study is not correct and I found a lack of recent bibliography and a poor
discussion.
Response - Thank you for the time to review our manuscript and for providing valuable comments and suggestions, which helped us to significantly improve its quality. We have taken your comment into account throughout the manuscript, ensuring that these future implications are clear. Thus, new phrases were introduced in the discussions (lines 250-256, 262-263,270-274, 295-296, 322-333).
Thank you again!
Reviewer 2 Report
COMMENT 18: In my opinion, this work cannot be presented as a study of the Boar semen microbiota, since the semen was cultured diluted in extenders with antibiotics.
Boar semen contamination: identification of bacterial strains and antimicrobial resistance profile
The authors have modified the Title, I agree
COMMENT 19: In Material and Methods affirm “a 200 μl aliquot of each semen sample was added to sterile micro-88 tubes containing Stuart transport medium” (lines 88-89). What implies that the pure semen is the one that is cultured, however, in the discussion and in the conclusion it is described that the semen has been cultured diluted in extender with antibiotics.
The authors have clarified this issue
COMMENT 20: (Lines 234-242) “If it is taken into account that these strains were isolated from the diluted semen, given that the diluent contained two antibiotics and the boars were clinically healthy, it can be hypothesized that the bacteriospermia found is the consequence of semen contamination during harvesting or processing semen processing laboratory. For this reason, we consider that it is particularly important to have a high degree of hygiene of the staff, equipment and laboratory where the semen is taken and processed. The production of doses of semen with low bacterial contamination and high viability of sperm will be possible only with a strict hygienic control in the processing of semen, and with the knowledge of the antibiotic resistance profile of bacteria that can contaminate semen.”
The authors have partially clarified this issue
New comments:
1.- The authors comment that “the study is carried out with semen samples were collected from boars clinically healthy, but reproductive failure manifested by genital and urinary tract infections, abortions, etc. were recorded in the farm”. However, in the study they decided to analyse only gram - bacteria, because gram – bacteria have the most harmful effect on the semen.
In my opinion, they should have analysed all microorganisms, as in most of the bibliography referred.
In addition to these gram-negative bacteria with known pathogenic action, non- pathogenic gram-positive bacteria were also identified - Streptococcus porcinus, Staphylococcus equorum, Staphylococcus succinus and Aerococcus viridans. These isolates were not further tested because it is known from the literature that gram-negative bacteria have the most harmful effect on the semen (lines 222-226)
Especially when the authors comment on the importance of semen contamination not only in the quality of the semen but also its possible pathogenicity in the reproductive tract of the sow.
The bacterial contamination of the sow reproductive tract by insemination artificial can cause metritis, endometritis, vulvar discarches, return to the oestrus, reduction of litter size, increased number of stillbirths and mummies. (lines 316-319)
In fact, the pathogenicity of some gram + bacteria in different pathological processes in the sow has been described. For example, some gram+ bacterias isolated in the study have been described in the following publications:
- Wang, Y., Guo, H., Bai, Y. et al. Isolation and characteristics of multi-drug resistant Streptococcus porcinus from the vaginal secretions of sow with endometritis. BMC Vet Res 16, 146 (2020). https://doi.org/10.1186/s12917-020-02365-9
A S. porcinus isolate with multi-drug resistance was identified from vaginal secretions of sows with endometritis in one pig breeding farm, which suggests that the sow endometritis was caused by S. porcinus infection during artificial insemination. This study indicates that sensitive antibiotics such as penicillin G or some cephalosporins could be used for treatment of the diseases. In addition, the study hints that bacterial multi-drug resistance is a tough problem for disease treatment in pig farms.
- Nguyen VG, Kim CU, Do HQ, Shin S, Jang KC, Park YH, Park BK, Chung HC. Characteristics of Aerococcus viridans isolated from porcine fetuses in Korean farms. Vet Med Sci. 2021 Jul;7(4):1325-1331. doi: 10.1002/vms3.456. Epub 2021 Feb 24. PMID: 33624943; PMCID: PMC8294361.
Swine abortion caused by viruses as well as bacteria has caused many economic losses in domestic farms over the years; however, bacterial abortion has not yet been studied in Korea. Several bacterial species were isolated from aborted fetuses (n = 103) for which the cause of death was not viral abortion. Among them, we focused on Aerococcus viridans, which had the highest positive rate within three provinces (Gangwon, Jeonnam and Gyeongnam). A total of 16 isolates were identified as A. viridans by matrix-assisted laser desorption ionization-time of flight mass spectrometry (MALDI-TOF MS), and 13 were characterized by both antibiotic resistance and 16S rRNA gene analysis. Based on antibiotic susceptibility testing result, eight antimicrobials could not effectively eliminate the present isolation (more than 40% of isolates can resist these antibiotics), while all except two strains were susceptible to trimethoprim/sulfamethoxazole. Molecular analysis indicated genetic variation among these strains. This study is the first report detecting A. viridans from aborted fetuses in Korean domestic farms.
2.- On several lines it is commented that gram- bacteria have only been found when the lines 222-226 specify the isolation of several gram + bacteria.
All isolates are gram-negative bacteria, and no any gram-positive bacteria were identified (line 219)
According to the data presented in Table 1 and Figure 1 several genera and bacterial species were isolated from the diluted seminal material, all of them being Gram negative bacteria (lines 267-268)
3.-The contamination of swine semen is a general concern that generates frequent publications, so it is important to compare the most recent publications in the discussion of the articles, a more recent and more in-depth bibliographic review on the subject has been missed.
For example:
Andressa C. Dalmutta,b, Luisa Z. Moreno b , Vasco T. M. Gomesb , Marcos P. V. Cunha c , Mikaela R. F. Barbosad , Maria Inês Z. Satod , Terezinha Knöblc , Antonio Carlos Pedrosoa and Andrea M. Moreno b Characterization of bacterial contaminants of boar semen: identification by MALDITOF mass spectrometry and antimicrobial susceptibility profiling JOURNAL OF APPLIED ANIMAL RESEARCH 2020, VOL. 48, NO. 1, 559–565 https://doi.org/10.1080/09712119.2020.1848845
4.- The discussion not contributes new information to the topic.
5.- I am not an expert in any English language, but basic inaccuracies of the language are appreciated
In conclusion, although the authors have managed to give a new focus to the article, the selection of the bacteria under study is not correct and I found a lack of recent bibliography and a poor discussion.
Author Response

(The authors gave the same response as above.)

Round 2
Reviewer 2 Report
Lines 3,4
The authors have modified the Title, again
LAST: Boar semen contamination: identification of bacterial strains and antimicrobial resistance profile
NEW: Boar semen contamination: identification of gram-negative bacteria and antimicrobial resistance profile
I am agree
Lines 17-18 The sentence should be rewritten
The aim of this study was to identify bacteria that appear in boar semen and to establish models of antimicrobial resistance of isolated bacteria
The aim of this study was to identify bacteria that appear in boar semen and to establish models of antimicrobial resistance of isolated gram-negative bacteria
They only study the antimicrobial resistance of gram-negative bacteria.
Lines 29-30 The sentence should be rewritten
This study focused on the identification of natural bacteria in wild boar semen and the impact on the quality of ejaculates obtained from wild boars, as well as the establishment of antimicrobial resistance patterns of isolated bacteria.
This study focused on the identification of natural bacteria in wild boar semen and the impact on the quality of ejaculates obtained from wild boars, as well as the establishment of antimicrobial resistance patterns of isolated gram-negative bacteria.
As they affirm the main source of this contamination is the boar, however, laboratory equipment, or distilled water used for semen extenders are also potential sources of contamination, and their study culture diluted semen with antibiotics.
They only study the antimicrobial resistance of gram-negative bacteria.
lines 73-78
The sentence should be rewritten
The objectives of present study focused on identification and microbiological examination of the boar sperm microbiota and analysis of the quality of these samples, using a rapid and precise working technique, the matrix-assisted laser desorption/ionization time-of-flight (MALDI-TOF) mass spectrometry and API 20 E (bioMerieux, France) for strains identification and further the establishment of the main antimicrobial resistance pathotypes.
The objectives of present study focused on identification and microbiological examination of the boar sperm microbiota doses and analysis of the quality of these samples, using a rapid and precise working technique, the matrix-assisted laser desorption/ionization time-of-flight (MALDI-TOF) mass spectrometry and API 20 E (bioMerieux, France) for strains identification and further the establishment of the main antimicrobial resistance pathotypes of isolates gram-negative bacterias.
Line96
The sentence should be rewritten and it would be useful to add the commercial name of the extender
All centers used the same extender which contain gentamicin as antibiotic.
All centers used the same extender (extender name) containing gentamicin as an antibiotic.
Line 126
The raw semen and the extender were bacteriologically tested in the boar centers laboratories.
The results are not described in the text.
Lines 195-196 The sentence should be withdrawn: “All isolates are gram- negative bacteria, and no gram-positive bacteria were identified”.
In lines 200 the authors indicate: “...gram-positive bacteria were also identified”
Lines 199-201
In addition to these gram-negative bacteria with known pathogenic action, non- pathogenic gram-positive bacteria were also identified - Streptococcus porcinus, Staphylococcus equorum, Staphylococcus succinus and Aerococcus viridans. These isolates were not further tested because it is known from the literature that gram-negative bacteria have the most harmful effect on the semen [3,4].
“non-pathogenic” should be withdrawn, there is bibliography that indicates otherwise,
For example, some gram+ bacterias isolated in the study have been described in the following publications:
- Wang, Y., Guo, H., Bai, Y. et al. Isolation and characteristics of multi-drug resistant Streptococcus porcinus from the vaginal secretions of sow with endometritis. BMC Vet Res 16, 146 (2020). https://doi.org/10.1186/s12917-020-02365-9
A S. porcinus isolate with multi-drug resistance was identified from vaginal secretions of sows with endometritis in one pig breeding farm, which suggests that the sow endometritis was caused by S. porcinus infection during artificial insemination. This study indicates that sensitive antibiotics such as penicillin G or some cephalosporins could be used for treatment of the diseases. In addition, the study hints that bacterial multi-drug resistance is a tough problem for disease treatment in pig farms.
- Nguyen VG, Kim CU, Do HQ, Shin S, Jang KC, Park YH, Park BK, Chung HC. Characteristics of Aerococcus viridans isolated from porcine fetuses in Korean farms. Vet Med Sci. 2021 Jul;7(4):1325-1331. doi: 10.1002/vms3.456. Epub 2021 Feb 24. PMID: 33624943; PMCID: PMC8294361.
Swine abortion caused by viruses as well as bacteria has caused many economic losses in domestic farms over the years; however, bacterial abortion has not yet been studied in Korea. Several bacterial species were isolated from aborted fetuses (n = 103) for which the cause of death was not viral abortion. Among them, we focused on Aerococcus viridans, which had the highest positive rate within three provinces (Gangwon, Jeonnam and Gyeongnam). A total of 16 isolates were identified as A. viridans by matrix-assisted laser desorption ionization-time of flight mass spectrometry (MALDI-TOF MS), and 13 were characterized by both antibiotic resistance and 16S rRNA gene analysis. Based on antibiotic susceptibility testing result, eight antimicrobials could not effectively eliminate the present isolation (more than 40% of isolates can resist these antibiotics), while all except two strains were susceptible to trimethoprim/sulfamethoxazole. Molecular analysis indicated genetic variation among these strains. This study is the first report detecting A. viridans from aborted fetuses in Korean domestic farms.
ADD “gram-negative” in all of the tables and figures
Line 206, Table 1. Synopsis of bacterial genera and species isolated from doses of diluted boar semen (DS) isolated in pure culture or in mixed culture.
Line 213, Figure 1. Proportion of bacteria isolated and identified in the doses of diluted semen (DS).
Line 218 Table 2. Behavior of isolated bacterial strains in antimicrobial substances.
Line 239:The sentence should be withdrawn “..., all of them being Gram negative bacteria”. It is the authors themselves who have obviated the analysis of gram-positive bacteria.
Line 241 The sentence should be rewritten
From 31 samples (67.39%) a single bacterial genus was isolated. In the other 15 samples (32.60%) two bacterial genera were identified.
From 31 samples (67.39%) a single gram-negative bacterial genus was isolated. In the other 15 samples (32.60%) two gram-negative bacterial genera were identified.
Lines 282-285
Ralstonia picketii and Achromobacter xylosoxidans can be found in the water distillation system of a boar stud facility that uses this water to expand raw semen. This study showed that the presence of A. xylosoxidans and R. pickettii in water for semen extension of porcine semen does not detrimentally affect sperm motility or pH of the final solution regardless the choice of semen diluent [16,18,24,25].
“This study showed that …”... it is understood that the authors themselves have determined this statement,when maybe they wanted to reference another study, the sentence should be rewritten for better understanding
Lines 303-307
If it is taken into account that these strains were isolated from the diluted semen, given that the diluent contained an antibiotic and the boars were clinically healthy, it can be hypothesized that the found bacteriospermia is the consequence of semen contamination during harvesting or diluted semen doses production.
This sentence is unclear,
Although the boars didn’t have clinical symptoms, it cannot be guaranteed that it does not have bacteriospermia, since many infections of the male reproductive tract do not imply clinical disease and are only visible when a poor seminal quality is perceived, reproductive indices decrease or it is appreciated that females present uterine infection, etc.
The authors indicated in line 103. “ All boars were clinically healthy, but on farms there were registered reproductive failures manifested by genital and urinary tract infections, abortions, etc”.
Furthermore, the authors have observed low motility at some doses (lines 169-170).
No alterations regarding sperm color and concentration were observed, but the presence in semen samples of E. coli, Burkholderia cepacia, Serratia marcescens and Proteus mirabilis was negatively associated with sperm motility (p < 0.05).
Según los autores la motilidad se realizó en semen sin diluir (line 119) lo que implica que la contaminación no proviene del diluyente
According to the authors, motility was performed in raw semen (line 119), which implies that, at least contamination with those bacteria, does not come from the extender or semen doses production.
For the sample analysis, Leja blades of 30 μl, 118 with 4 chambers (Cryo BioSystem, France) were used, in which 10 μl semen was placed, 119
Equally unclear is to say that “since the extender has antibiotics, the contamination comes from the handling, or from the preparation of the doses”, since the addition of antibiotics to the diluent is intended to prevent bacterial growth from handling and processing.
I agree that the bacterial contamination of seminal doses is very likely to come from handling, etc... but not for the reasons stated here.
I believe that the authors' appreciation is correct when referring in this case to the antimicrobial resistance of the gram-negative bacteria found, to the antibiotics used in the diluent used.
If the paragraph is rewritten, the following considerations should also be taken into account.
Line 303 …….. that these gram-negative strains were isolated from the diluted semen...
Line 305 …. “Bacteriospermia” It cannot be considered that when we refer to bacterial contamination in semen doses.
Line 315 The sentence should be rewritten
In this study, all isolated gram-negative bacterial strains presented resistance phenotype against at least one of the tested antimicrobial groups
Line 328 The sentence should be rewritten
Most of the gram-negative bacteria isolated in our study had a high level resistance rate of …..
Lines 338-339
In this study, following the bacteriological examinations of the doses of diluted semen, 36.45% of them were positive, I have not found the following percentage, 36.45% in the text, in fact the authors indicate in line 186 “Out of the 96 tested doses of diluted semen (DS), only 46 (47.91%) were positive at 186 bacteriological exams”.
Line 340 “...which means a high bacteriemia, ..”
“Bacteriemia” cannot be considered when we refer to bacterial contamination in semen doses..
Lines 342-344
Understanding bacterial dynamics during microbial contamination, identification of the main contaminating bacterial genera and their origin, as well as behavior to antimicrobials is mandatory for proper quality control and to reduce the risk of contamination of semen doses.
The authors have not studied the dynamics during microbial contamination, nor the origin of the contamination. It would have been very interesting to include it in the experimental design.
Lines 348
…...microbiological screening of bacteriospermia in boars …..
“Bacteriospermia” cannot be considered when we refer to bacterial contamination in semen doses.
General comments
In the discussion and in the conclusion some indications to help to reduce the bacterial load of seminal doses are in lack
I am not an expert in any English language, but basic inaccuracies of the language are appreciated
p.e. The bacterial contamination of the sow reproductive tract by insemination artificial can cause metritis
Bacterial contamination of the sow's reproductive tract by artificial insemination can cause metritis,...
p.e homogenize throughout the text the word “extender”, instead of diluent
p.e homogenize throughout the text the word “gram-negative”, instead of Gram negative (ine 334)
Format errors, spaces between lines and between words are appreciated

Author Response
Please see the attachment

This manuscript is a resubmission of an earlier submission. The following is a list of the peer review reports and author responses from that submission.
Round 1
Reviewer 1 Report
Review Manuscript ID: animals-1425121, entitled ‘Boar semen microbiota: identification of bacterial strains and antimicrobial resistance profile’
The aim of the study was to identify and examine the boar sperm microbiota and to analyze its quality.
The Authors highlight an undoubtedly important aspect in assessing the quality of boar semen used in insemination. There are great ambiguities in methodology and the research is poorly planned. The Authors write the semen samples came from three artifical insemination centers. I suppose it was bad solution to use the semen from three different centers and not to take into consideration the statistical analysis of the variability resulting from the influence of the insemination station. It would be better to analyze the semen from one insemination center.
The authors do not state how many boars were from each station and what were the breeds. How were sperm cells parameters evaluated and were they normal or not?
The References section should be adapted to the Animals requirements.
Reviewer 2 Report
The question asked by this paper is an interesting one, but the paper presentation needs extensive revisions.
The whole paper needs to be revised for English style and clarity. Suggested revisions for the simple summary and abstract are included here, but the rest of the paper was not revised for English style. The simple summary is too long, and most of what is written should be reserved for the abstract or introduction.
Simple Summary: The boar seminal material semen can contain many bacterial species, some of them be-14 ing a of which can have a negative impact upon the quality of the semen, as well as on the sows’ reproductive capacity. The main negative effect can be seen in a decreased number of piglets / sows, repeated heat cycles, and mortalities in sucklings. The semen contamination may occur at time of collection,when the sperm is collected from 17 boar, or during semen processing later, in the processing of the sperm in laboratory. The aim of this study was to identify the bacteria that appear in boar semen the seminal material during the processing semen doses, and to establish models of antimicrobial resistance of isolated bacteria. It is very important to know which antimicrobials are effective efficacy or not, because the semen doses are diluted and this diluent contain some anti-21 microbials. The results obtained show that the semen doses examined contained different bacterial species, some of them with a known negative effect on sows’ reproductive tract (Pseudomonas, Enterobacter, Klebsiella, E. coli). Unfortunately, and more than half of these isolates were resistance at to gentamycin (56.52%) and penicillin (58.69%), antimicrobials commonly used in boar semen extenders. CONCLUSION SENTENCE?
Abstract: Presence of bacterial contamination in pigs’ seminal doses of boar semen occurs with some frequency in artificial insemination centers, and may have a negative effect on the quality of the semen as well as on the sows’ reproductive capacity. The main source of this contamination is the own boar, however, laboratory equipment, or distilled water which are used in laboratory used for semen extenders are also potential sources of contamination. This study focused on the identification of naturally occurring bacteria in seminal material boar semen, and impact on the quality of ejaculates obtained from boar, as well as on the establishment of antimicrobial resistance patterns of isolated bacteria. Semen samples were collected from 96 boars, ranging in age from 12-36 month, from three artificial insemination centers from the North-West of Romania. Bacterial species were identified by two methods: matrix-assisted laser desorption/ionization time-of-flight (MALDI-TOF) mass spectrometry and API 20 E (bioMerieux, France). The results obtained show that the main bacteria isolated from the doses of seminal material semen are represented by the were gram-negative bacteria (47.91%), with a majority of the contaminant bacteria belonging to the family Enterobacteriaceae: Seratia marcescens 19.56%, Proteus mirabilis 15.21% and Escherichia coli 10.86% and to the family Pseudomonaceae: Ralstonia picketii 17.39%, Burkholderia cepacia 10.86%, Pseudomonas aeruginosa 8.69% and Pseudomonas fluorescens 4.34 % respectively. More than half of these isolates (56.52%) were resistanct to e at gentamycin and 58.69% were resistance at t to penicillin. These antibiotics are very frequently added in sperm diluent in the centers for the processing of sperm from boars in Romania. CONCLUSION??
Other suggestions for the paper:
Line 87: Please clarify what you mean by sanitization by manual pressure.
Material and Methods: Please explain what the standards were for color assessment, and what the settings were for CASA evaluation. Semen culture needs more explanation- how many times were the samples examined, and at what specific intervals? What culture media was used, and what were the diluent concentrations?
Lines 100-103 are part of submission instructions.
Please expand your description of statistical methods.
Please use appropriate subheading format.
RESULTS
I am concerned about the accuracy of the results given the failure to isolate any gram positive bacteria, which are commonly found in other studies looking at boar semen contamination. Was this because the diluent had an antibiotic in it?
Can you please include motility, concentration and morphology results?
Do you have pregnancy rates for the boars included in the study?
Discussion
I don't think the first sentence is accurate- not all boar semen/ reproductive tracts are free of bacteria!
The discussion needs editing for clarity and format.
Reviewer 3 Report
In my opinion, this work cannot be presented as a study of the Boar semen microbiota, since the semen was cultured diluted in extenders with antibiotics.
In Material and Methods affirm “a 200 μl aliquot of each semen sample was added to sterile micro-88 tubes containing Stuart transport medium” (lines 88-89). What implies that the pure semen is the one that is cultured, however, in the discussion and in the conclusion it is described that the semen has been cultured diluted in extender with antibiotic
(Lines 234-242) “If it is taken into account that these strains were isolated from the diluted semen, given that the diluent contained two antibiotics and the boars were clinically healthy, it can be hypothesized that the bacteriospermia found is the consequence of semen contamination during harvesting or processing semen processing laboratory. For this reason, we consider that it is particularly important to have a high degree of hygiene of the staff, equipment and laboratory where the semen is taken and processed. The production of doses of semen with low bacterial contamination and high viability of sperm will be possible only with a strict hygienic control in the processing of semen, and with the knowledge of the antibiotic resistance profile of bacteria that can contaminate semen.”
The authors hypothesize that the contamination of the semen has been during processing. However:
- Processing is not described in material and methods
- There are no sample controls, no pure semen cultured, and no extender cultured without semen.